# MULTISENSORY GEOSPATIAL MODELS VIA CROSS-SENSOR PRETRAINING

## ABSTRACT

Geospatial remote sensors, derived from optical and microwave sensors, exhibit significant diversity and provide unique capabilities due to the different observing mechanisms. By integrating multi-sensor data through fusion, researchers can harness the complementary and synergistic nature of optical and microwave data to achieve more accurate and efficient Earth monitoring. Despite the proven enhancements by geospatial pretraining models on various downstream tasks, most research primarily focuses on single sensor modality. Thus, to unlock these synergies, we introduce a multi-sensor geospatial pretraining model, XGeo, pretrained with four sensor modalities: RGB channels, Sentinel-2, Synthetic Aperture Radar (SAR), and Digital Surface Model (DSM) data, encompassing a total of two million multi-sensor images. Our method is equipped to manage both paired and unpaired data effectively. When originating from the same geolocation, we integrate cross-linked corresponding sensors into the modeling of the masked image, which facilitates the learning of a joint representation from multiple sensors. In addition, we utilize a mixture of expert layers and heterogeneous batches to mitigate data heterogeneity. Our experiments show that XGeo enhances performance on both single sensor and multisensor downstream tasks, such as land-use classification, segmentation, cloud removal, and pan-sharpening. We also reveal that representations from natural images differ from some of geospatial remote sensors, which renders the use of existing representations less effective. Our work serves as a comprehensive guide for developing robust multisensor geospatial pretraining models, paving the way for more advanced geospatial capabilities.

## 1 INTRODUCTION

Geospatial remote sensors exhibit considerable diversity (Figure 1), with reported spatial (Qiu et al., 2013) and feature heterogeneity (Vanderhoof et al., 2023; Xu et al., 2022a). Two principal categories emerge based on their imaging mechanisms: optical sensors (e.g., Sentinel-2 (Drusch et al., 2012) and LiDAR) and microwave sensors (e.g., Synthetic-aperture radar (Fornaro & Pascazio, 2014)). These sensors vary significantly in their observation methods and capabilities. Optical remote sensing captures reflected and absorbed electromagnetic radiation in the visible and near-infrared spectrum, yielding high-resolution imagery and surface property information. Conversely, microwave remote sensing operates at longer wavelengths, penetrating clouds and vegetation to reveal subsurface features and structural properties (Musa et al., 2015) (Figure 1).

A multi-sensor fusion approach combines the strengths of both optical and microwave remote sensing, offering a more comprehensive and accurate understanding of the Earth's surface (Schmitt et al., 2017). By integrating data from multiple sensors, researchers can leverage the complementary nature of optical and microwave data to overcome limitations and obtain a more complete picture. For instance, combining optical and microwave data can help estimate soil moisture content, which is crucial for ecosystem management (Jung et al., 2017; Gottfriedsen et al., 2021). Multi-sensor fusion also enhances the accuracy of topographic mapping by incorporating both surface features captured by optical sensors and elevation information derived from microwave sensors. Numerous multisensor fusion deep learning models have been proposed for individual tasks, such as cloud removal (Xu et al., 2022b; He et al., 2021b), biomass estimation (Ghosh & Behera, 2018) and landcover segmentation Cha et al. (2021); Hu et al. (2023). These studies substantiate the enhancement in performance achievable by geospatial models incorporating multisensor modalities.

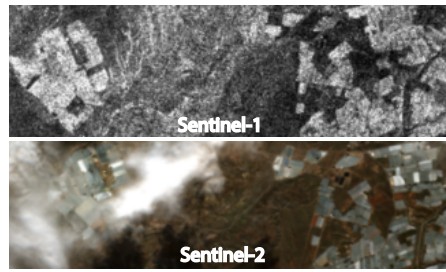 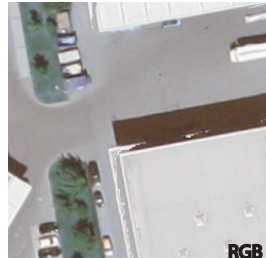 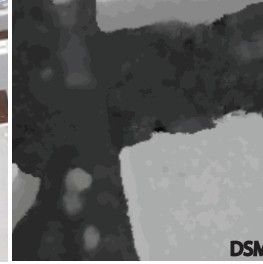

Figure 1: Examples of four sensor modalities: SAR, Sentinel 2, RGB, DSM. Here, each pair of {SAR & Sentinel-2} and {RGB & DSM} are colocated on the same geolocation respectively. In the example of Sentinel-2, only blue, green, red bands are shown for the convenience of visualization.

Despite these important synergies, most of geospatial pretrained models focus predominantly on a single modality (Mendieta et al., 2023; Wang et al., 2022a; Cong et al., 2022; Mañas et al., 2021; Sun et al., 2022). While studies like Liu et al. (2022a), Chen & Bruzzone (2022) and Scheibenreif et al. (2022) employ Sentinel-2 and SAR for pretraining via contrastive learning, these methodologies are inherently limited by the need to paired sensor modalities. This limitation restricts the efficient utilization of the abundant unpaired sensor modalities that are prevalently available in real-world scenarios. By establishing a multisensor pre-trained model scalable to both paired and unpaired sensors, a unified framework for analyzing multisensor remote sensing data can be provided. Such a model can be fine-tuned or used as a feature extractor to interpret multisensor data effectively.

Therefore, our paper develops a novel multi-sensor geospatial pretraining model that can potentially be scalable to many sensor modalities, paired or not. To our best knowledge, it is the first multi-sensor geospaital pretraining of such kind. Additionally, our paper seeks to address several unexplored questions in the realm of multisensor geospatial models. A natural inquiry arises: *How can joint representations between corresponding sensors be learned by employing masked image modeling techniques?* In geospatial tasks within the RGB domain, it is typical to leverage pre-trained backbones on ImageNet (Risojevic & Stojnic, 2021; Wang et al., 2022b) or to utilize features distilled from such models (Mendieta et al., 2023). Given this, we inquire, *Does leveraging or distilling features from established vision models enhance multisensor geospatial pretraining?* Lastly, a practical concern emerges: *How can multisensor heterogeneity be mitigated during pretraining?* In this paper, we aim to address the aforementioned questions and provide insights into effectively pretraining multisensor geospatial models. Our contributions can be summarized as follows:

- We introduce a novel cross-sensor paradigm, XGeo, for joint representation learning. This paradigm harmonizes diverse representations and empowers multisensor models to effectively discern the complex relationships between corresponding sensors.

- We unveil a high-performing pretrained model, cultivated from a comprehensive multisensor dataset, XGeoSet, encompassing over 2 million images. This model adeptly amalgamates four sensor modalities: RGB images, multispectral images, SAR, and DSM, demonstrating superior performance across several multisensor downstream tasks.

- Furthermore, we have made discoveries, yet to be reported, that several methods can augment the model's performance: (1) The incorporation of MoE and heterogeneous batching effectively harmonizes diverse representations from optical and microwave sensor types; (2) Initiating pretraining from scratch has been observed to yield superior results compared to leveraging existing foundational models.

## 2 RELATED WORK

**Geospatial pretraining.** As pretrained models continue to revolutionize the fields of vision and natural language processing, their potential in the geospatial sphere is becoming increasingly evident. These models have demonstrated remarkable prowess in enhancing the efficacy of deep learning models when applied to downstream tasks (Neumann et al., 2019; Mañas et al., 2021; Cong et al., 2022; Ayush et al., 2020; Mendieta et al., 2023). The geospatial domain has seen the emergence of two main approaches for self-supervised pretraining paradigms. The first centers around the

use of contrastive learning (Mañas et al., 2021; Ayush et al., 2020; Liu et al., 2022a). In this technique, the loss function is crafted to incentivize the model to draw similar or positive pairs closer together in the embedding space while pushing dissimilar or negative pairs further apart (Chen et al., 2020). However, identifying appropriate augmentations for contrastive methods presents a significant challenge. Certain augmentations in geospatial images, which significantly alter the image's intensity, can lead to undesirable outcomes (Neumann et al., 2019). Various implementations of pretraining with contrastive learning incorporate temporal and spectral augmentation (Mañas et al., 2021), while others apply a colorization objective (Vincenzi et al., 2020). Although works such as Liu et al. (2022a), Chen & Bruzzone (2022) and Scheibenreif et al. (2022) treat colocalized Sentinel-2 and SAR as positive pairs, these approaches are restricted to these two or more pairing sensor modalities and doesn't efficiently leverage the wide range of unpaired sensor modalities. Given these augmentation constraints (Neumann et al., 2019), alternative methods have been developed, employing Masked Image Modeling (MIM) (Cong et al., 2022; Mendieta et al., 2023; Sun et al., 2022), relying on simple spatial augmentations such as flipping and cropping. MIM not only requires less stringent augmentations but also outperforms its contrastive learning counterparts (Mendieta et al., 2023; Cong et al., 2022; Sun et al., 2022). However, most prior studies focus on remote sensing imagery in the visible spectrum or employ a single sensor modality (Mendieta et al., 2023; Wang et al., 2022a; Cong et al., 2022; Mañas et al., 2021). Alternatively, they are confined to *two or more paired* sensors due to the inherent limitations of contrastive learning (Liu et al., 2022a; Chen & Bruzzone, 2022; Scheibenreif et al., 2022). In this work, we develop our pretraining objective based on a masked image modeling approach, akin to (Xie et al., 2021; He et al., 2021a). We demonstrate that our model can be pretrained with four sensor modalities, taking advantage of the unpaired sensor.

**Multi-source learning in language and vision communities.** Multi-source learning is a prevalent strategy when handling multi-modal tasks (Shen et al., 2023; Zhu et al., 2022) or multitask challenges in both the language and vision domains (Chen et al., 2022; Aoki et al., 2022; Liang et al., 2022; Aghajanyan et al., 2021; Lample & Conneau, 2019; Conneau et al., 2020; Bachmann et al., 2022). This technique exploits data from diverse sources to bolster the learning process and enhance model performance. A notable example is multilingual pretrained models, such as XLM (Lample & Conneau, 2019) and its derivatives (Conneau et al., 2020; Chi et al., 2022). These models utilize multilingual datasets, pretraining them on a large scale to generate unsupervised cross-lingual representations (Conneau et al., 2020; Lample & Conneau, 2019). This approach enables the models to develop a unified representation across multiple languages, thereby enhancing their performance on cross-lingual tasks. Furthermore, the batching strategy has been identified as an essential aspect of creating generalizable representations and preventing collapse in multilingual models (Aghajanyan et al., 2021; 2020). Simultaneously, the Mixture-of-Experts (MoE) strategy (Shazeer et al., 2017) has been utilized to enhance multi-source learning in both multitask learning (Chen et al., 2022; Aoki et al., 2022; Liang et al., 2022) and language-vision pretraining (Shen et al., 2023; Zhu et al., 2022). In the specific context of multisensor geospatial pretraining, heterogeneity can originate from the use of different sensor types (e.g., optical, microwave) or different platforms (e.g., various satellites). Properly addressing this heterogeneity is crucial as it can significantly influence the performance of the pretraining model (Aghajanyan et al., 2021). To meet this challenge, we draw inspiration from works in computer vision (Riquelme et al., 2021) and natural language processing (Aghajanyan et al., 2021; Lample & Conneau, 2019). We incorporate techniques such as cross-sensor representation learning into our XGeo.

# 3 CROSS-SENSOR GEOSPATIAL PRETAINING

In this section, we present the multisensor pretraining paradigm. Following (Mendieta et al., 2023), we employ SIMMIM (Xie et al., 2021) using a Swin Transformer (Liu et al., 2021; 2022b) as a backbone. Figure 2 presents an overview of the cross-sensor geospatial pretraining methodology. .

## 3.1 INPUT REPRESENTATION

**Distinct embedding layers for each sensor.** We consider $N$ sensor modalities, with each sensor having a corresponding number of channels, denoted as $\{C_i\}_{i=1...N}$. Taking into account the unique number of channels associated with each sensor (some examples shown in Table 1), we utilize individual patch embeddings tailored to each specific sensor. This approach allows the model to efficiently process and learn from the distinct characteristics of various sensor modalities.

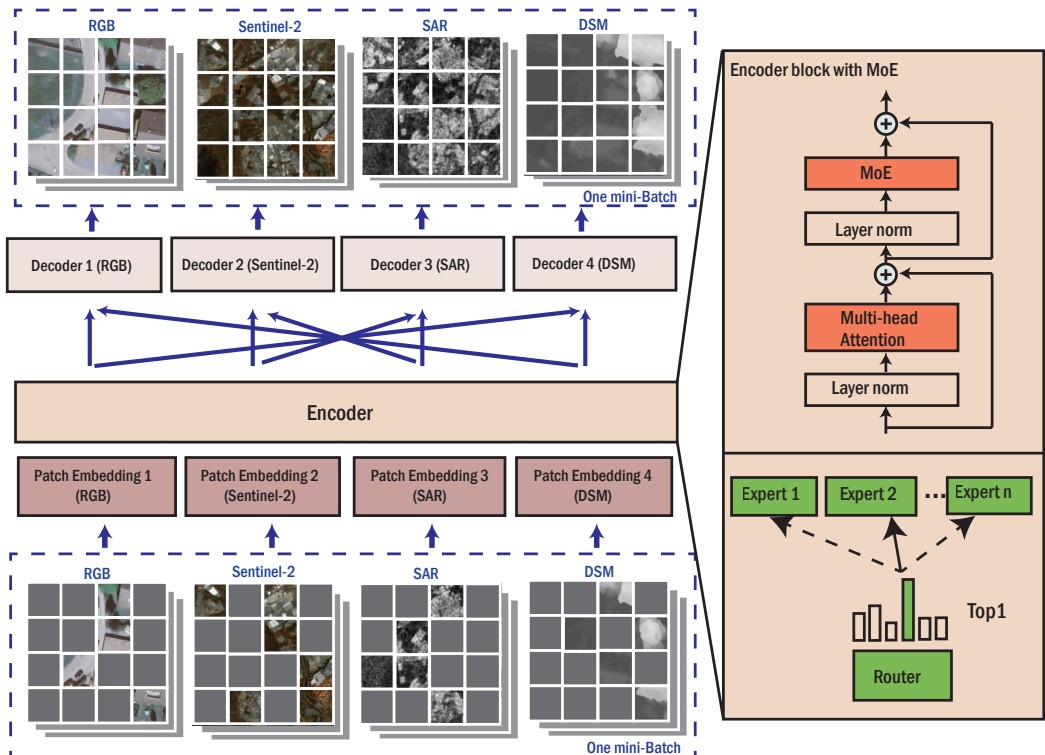

Figure 2: Overview diagram of XGeo. Each sensor is fed through a separate patch embedding layers (Section 3.1) and through the same encoder. During the reconstruction, separate decoders are used. If the sensors are paired, there's a chance that our model will reconstruct the corresponding paired sensor instead itself (Section 3.2). Other best practice can be found at Section 3.4.

To elaborate, for the image from the $i$-th sensor, $\boldsymbol{I} \in \mathbb{R}^{W \times H \times C_i}$, we first divide it into square patches of size $P$, resulting in $\boldsymbol{T}_i \in \mathbb{R}^{L \times P^2 C_i}$. Here, $W$ and $H$ represent the width and height of the images, while $L$ denotes the number of patches. We then apply corresponding linear embedding layers, $\{f_i\}_{i=1...N} : \mathbb{R}^{L \times P^2 C_i} \to \mathbb{R}^{L \times C_e}$, to the patches $\boldsymbol{T}$, projecting them to an embedding space with the dimension of $C_e$. The function $f_i$ represents the embedding layer for images from the $i$-th sensor. In our work, $C_e$ is the same for all sensors. Intuitively, to address the channel heterogeneity among various sensor modalities, each sensor modality will pass through a separate trainable embedding layer to *unify* the representation dimension before being fed into the shared encoder.

## 3.2 CROSS-SENSOR PRETRAINING

**Shared encoder for all sensor modalities.** The patches obtained from $\{f_i\}_{i=1...N}$, $\boldsymbol{T}_i \in \mathbb{R}^{L \times C_e}$, will then be masked and fed through the encoder. The masking strategy employed in our approach are the same as those used in Xie et al. (2021). By having separate patch embedding layers ($f_i$) for each sensor, the model can learn the unique characteristics of each sensor modality. The learned embeddings from all sensors are then integrated through the same encoder, enabling the model to effectively learn joint representations and handle multisensor geospatial data.

**Separate decoder for each sensors and cross sensor prediction.** Collecting data from different sensors for the same geo-location is a common practice in the geospatial domain. Learning joint representations of such multisensor data can prove beneficial for various downstream tasks. Although contrastive learning has demonstrated promise in learning effective representations, its performance may be limited due to the lack of suitable data augmentations for remote sensing images. To address this issue, we propose employing cross-sensor strategies in the context of MIM to learn joint representations for multisensor geospatial data. For instance, when the model is fed with masked images from DSM, it can predict the masked patches of itself or the corresponding images from

RGB. An example pair of DSM and RGB images is shown in Figure 1 in the two panels on the right. This encourages the model to align the different sensor representations. Accordingly, our model incorporates different decoders for each sensor.

Specifically, if there exists a pair of images from the $i$-th sensor and $j$-th sensor, $\{I_i \in \mathbb{R}^{W \times H \times C_i}, I_j \in \mathbb{R}^{W \times H \times C_j}\}$, the model processes the masked image as follows:

$$I'_i = D_i(En(f_i(I_i))) \text{ , or } I'_j = D_j(En(f_i(I_i)))$$
$$\text{and } I'_j = D_j(En(f_j(I_j))) \text{ , or } I'_i = D_i(En(f_j(I_j))) \tag{1}$$

where $En : \mathbb{R}^{W \times H \times C_i} \to \mathbb{R}^{L \times C_m}$ is the shared encoder, and $C_m$ is the embedding dimension of the final layer in the encoder. $I'_i$ and $I'_j$ are the predicted $i$-th and $j$-th sensor images respectively. $D_i : \mathbb{R}^{L \times C_m} \to \mathbb{R}^{W \times H \times C_i}$ is the decoder to reconstruct the $i$-th sensor image. Equation 1 shows that the predicted output of the pretraining model will either reconstruct itself or its paired sensor images.

If there's no paired sensor in the pretraining dataset, it will construct itself in the conventional way:

$$I'_i = D_i(En(f_i(I_i))) \tag{2}$$

This approach capitalizes on the inherent relationship between different sensors observing the same location, enabling the model to capture complementary information. Furthermore, it provides flexibility in handling scenarios where no paired sensors are available, allowing for enhanced adaptability in choosing the pretraining dataset. This is particularly advantageous given that multimodal geospatial datasets are less prevalent than single-sensor dataset.

## 3.3 PRETRAINING DATA.

Table 1: Breakdown of datasets of our pretraining data. We gather approximately 2M samples from a combination of labeled and unlabeled satellite imagery with various ground sample distances and sensor modalities.

| Dataset | # Images | GSD | Sensor modality | # Channels | paired sensors? |
|---|---|---|---|---|---|
| GeoPile (Mendieta et al., 2023) | 600K | 0.1m - 30m | RGB[a] | 3 | ✗ |
| MillionAID (Long et al., 2021) | 1M | 0.5 - 153m | RGB[a] | 3 | ✗ |
| SEN12MS (Schmitt et al., 2019) | 320K | 10m | SAR / sentinel-2 | 2/14 | ✓ |
| MDAS (Hu et al., 2023) | 40K | 0.1m - 10m | DSM/ RGB[b] | 1/3 | ✓ |

[a] is not sourced from a single sensor; instead, it amalgamates sensor images from an array of satellites, including NAIP, GeoEye, WorldView, QuickBird, IKONOS, and SPOT satellites, among others. [b] is derived from airborne sources (Hu et al., 2023). For more in-depth details regarding the RGB, please refer to Appendix A.2

Our multisensor pretraining data, XGeoSet, is composed of four sensor modalities, amassed to a total of 2 million images through the inclusion of additional geospatial data. The detailed composition of XGeoSet is presented in Table 1. Specifically, XGeoSet incorporates SEN12MS (Schmitt et al., 2019), a dataset enriched with paired SAR and Sentinel-2 satellite images from all meteorological seasons, to augment data diversity. All sensors in this dataset are ortho-rectified (Schmitt et al., 2019). Additionally, we have integrated DSM and RGB images from the MDAS dataset (Hu et al., 2023), resized to a dimension of 384.

It is noteworthy that, although the Sentinel-2 modality does encompass RGB channels in terms of imaging band, this RGB modality is distinguished and separated due to its expansive dataset that extends beyond the Sentinel-2 sensor, exhibiting varied Ground Sample Distances (GSD) and high feature entropy. Those two attributes that have been validated as impactful during pretraining (Mendieta et al., 2023; Cong et al., 2022). The exclusion of the RGB modality from our pretraining dataset results in a decrease in efficacy compared to datasets where it is included (Appendix A.2). The enhanced version of XGeoSet with RGB (GeoPile (Mendieta et al., 2023) and MillionAID (Long et al., 2021)) demonstrates elevated performance in the 7 downstream tasks outlined in GFM (Mendieta et al., 2023) under identical settings (Refer to Appendix A.1). To optimize XGeoSet-RGB, experimentation with diverse datasets was undertaken (See Appendix A.1).

### 3.4 BEST PRACTICE IN PRETRAINING

**MoE for multisensor learning** A shared encoder can present challenges when it comes to efficiently learning each sensor's representation. To tackle this issue, we propose integrating the sparsely gated Mixture of Experts (MoE) approach (Shazeer et al., 2017) to replace MLP layers within the encoder. Our pretraining loss function, $L$, combines L1 loss (Xie et al., 2021; He et al., 2021a) for reconstruction (i.e., MIM loss) and auxiliary losses (Hwang et al., 2022; Riquelme et al., 2021): $L = L_{\text{MIM}} + \lambda L_{\text{auxiliary}}$, where $\lambda$ represents the weight for auxiliary losses. In practice, we use $\lambda = 0.01$.

**Pretraining Method.** We utilize *heterogeneous batches* during the pretraining process, a method initially introduced by Muppet (Aghajanyan et al., 2021). This method is typically employed in multitask learning scenarios, aimed at creating a consolidated representation across diverse tasks during model training, notwithstanding their differing learning objectives. Our findings suggest that this method can also be effectively applied to multisensor geospatial pretraining (Section **??**). In each batch, all sensor data are loaded in sequence, ensuring that the optimization of our model encompasses all tasks (Aghajanyan et al., 2021). To elaborate, each batch can be depicted as a set: $\{ \boldsymbol{I} \in \mathbb{R}^{W \times H \times C_i} \}_{i=1\ldots N}$. Moreover, in light of the distinct imaging mechanisms inherent to these sensors, we opt to perform pretraining *from scratch* (Section 4.3).

## 4 EXPERIMENTS

**Experimental Settings.** All of our experiments are conducted using a SwinV2-base architecture (Liu et al., 2022b) with a patch size of 16×16 pixels and 8 experts. The models are pre-trained for either 100 epochs for ablation studies or 800 epochs to achieve optimal results and maintain comparability with state-of-the-art methods. When specified, 1% BEN and 1% SEN12MS-CR are also employed for ablation studies. We utilize 8 NVIDIA V100 GPUs with a batch size of 2048 (128 per GPU) and an image size of 192×192. All pretraining settings follow the configurations described in (Mendieta et al., 2023). Detailed pretraining settings and pretraining reconstruction visualization can be found at Appendix B.1 and Appendix B.2 respectively.

**Downstream Evaluation.** Upon completion of the pretraining, we fine-tune and assess the model on a diverse range of downstream multisensor datasets. This aims to provide a comprehensive understanding of the model's performance potential across various tasks (refer to Section 4.1 for task details). Table 2 provides an overview of the downstream evaluation tasks, together with their respective sensor modalities. Among these tasks, the use of multisensor data can enhance the performance of land classification and segmentation. Meanwhile, cloud removal is inherently dependent on multisensor modalities and cannot be effectively tackled without them. Although pansharpening requires one optical sensor, it relies heavily on multi-spectral images. Detailed settings in those downstream tasks can be found in Appendix C.2.

Table 2: Downstream tasks. It covers various use cases in geospatial domain, with a range of ground sample distances and sensor modalities.

| Dataset | # Application | GSD | Sensor modality | # Channels |
|---|---|---|---|---|
| Big Earth Net (Sumbul et al., 2019) | Scenes classification | 10m - 60m | SAR / Sentinel-2 | 2/14 |
| Vaihingen (Rottensteiner et al., 2012) | Land segmentation | 0.09m | DSM / RGB | 1/3 |
| SEN12MS-CR (Ebel et al., 2020) | cloud removal | 10 - 60m | SAR / Sentinel-2 | 2/14 |
| SpaceNet | Pan-sharpening | 0.1m - 10m | WorldView 3 | 8 |

### 4.1 GEOSPATIAL DOWNSTREAM EVALUATION

**Scene classification.** One prevalent remote sensing application is classification. We evaluate our pretraining model on BigEarthNet (BEN) (Sumbul et al., 2019), a dataset extensively used in other literature (Mañas et al., 2021; Cong et al., 2022; Chen et al., 2020; Wanyan et al., 2023; Mendieta et al., 2023). BEN (Sumbul et al., 2019) is a large-scale remote sensing dataset specifically designed for multi-label classification tasks. The data includes pairs of 12-band Sentinel-2 images along with their corresponding 2-band SAR images. We employ the data split and 19-class evaluation, as is

Table 3: Quantitative results of all the downstream tasks (Table 2) from XGeo (ours) compared to other pretrained models. Results are replicated from the previous reports.

| Methods | 10% BEN mAP (↑) | 100% BEN mAP (↑) | SEN12MS-CR | | | SpaceNet | | Vaihingen mIOU (↑) |
|---|---|---|---|---|---|---|---|---|
| | | | MAE (↓) | SAM (↓) | SSIM (↑) | PSNR (↑) | SSIM (↑) | |
| SeCo (Mañas et al., 2021) | 82.6 | 87.8 | - | - | - | - | - | 68.9 |
| SatMAE (Cong et al., 2022) | 82.1 | - | - | - | - | 22.742 | 0.621 | 70.6 |
| MoCoV2 (Chen et al., 2020) | - | 89.3 | - | - | - | - | - | - |
| DINO-MC (Wanyan et al., 2023) | 84.2 | 88.6 | - | - | - | - | - | - |
| GFM (Mendieta et al., 2023) | 86.3 | - | - | - | - | 22.599 | 0.638 | 75.2 |
| Random | 82.6 | 86.2 | 0.048 | 14.78 | 0.572 | 21.825 | 0.594 | 67.0 |
| IN-22k (Liu et al., 2022b) | 85.7 | 89.5 | - | - | - | 21.655 | 0.612 | 74.7 |
| XGeo | **87.5** | **92.9** | **0.026** | **4.87** | **0.842** | **22.850** | **0.668** | **75.8** |

standard in the literature (Neumann et al., 2019; Mañas et al., 2021; Cong et al., 2022; Mendieta et al., 2023). In Table 3, we report the mean average precision (mAP) results on BEN for all methods. Our model can provide robust performance against other pretraining methods (Mañas et al., 2021; Cong et al., 2022; Chen et al., 2020; Wanyan et al., 2023; Mendieta et al., 2023; Liu et al., 2022b), including ImageNet-22k (Liu et al., 2022b). We note that those competing methods use different backbones. Their results of corresponding random initialization and ImageNet initialization can found in previous studies (Mendieta et al., 2023; Cong et al., 2022). Furthermore, one key motivation for training a geospatial foundation model is to improve the sample efficiency for downstream tasks. Notably, we find that our model maintains strong performance on BEN, even when only given 1% of the training data (Appendix C.3).

**Cloud removal** The majority of optical observations acquired via spaceborne Earth imagery are affected by clouds, presenting challenges in reconstructing cloud-covered information in previous studies. While optical imagery is impacted by adverse weather conditions

Table 4: Quantitative results of cloud removal, compared to existing models that are specially designed for cloud removal. Results are replicated from the original paper.

| Methods | MAE (↓) | SAM (↓) | SSIM (↑) |
|---|---|---|---|
| SAR-Opt-cGAN (Grohnfeldt et al., 2018) | 0.043 | 15.49 | 0.764 |
| DSen2-CR (Meraner et al., 2020) | 0.031 | 9.47 | 0.874 |
| GLF-CR (Xu et al., 2022b) | 0.027 | 7.65 | **0.885** |
| XGeo | **0.026** | **4.87** | 0.842 |

and lack of daylight, SAR sensors are not affected, providing a valuable source of complementary information. Consequently, performing cloud removal tasks without SAR data can significantly degrade task performance (Xu et al., 2022b). We evaluate our model on SEN12MS-CR (Ebel et al., 2020). Table 4 shows promising results in Spectral Angle Mapper (SAM) and Mean Absolute Error (MAE), outperforming existing cloud removal models (Grohnfeldt et al., 2018; Meraner et al., 2020; Xu et al., 2022b). Qualitative results are presented at Appendix C.5. Meanwhile, the structural similarity index measure (SSIM) metric demonstrates comparable results with those methods. Our multisensor pretraining approach, which incorporates SAR data, enables effective cloud removal, while other geospatial pretraining models that rely solely on optical data do not demonstrate their capabilities in cloud removal (Table 3).

**Pan-sharpening and Segmentation.** We extended our evaluation to include tests on pan-sharpening and segmentation. The results presented in Table 3 demonstrate that XGeo surpasses other models in performance. A more comprehensive discussion on these evaluations can be found in Appendix C.4.

## 4.2 COMPARISON WITH SINGLE SENSOR PRETRAINING.

To underscore the pivotal role of multiple sensor modalities in pretraining, we compare our multisensor pretraining approach using XGeoSet with models pretrained on only one sensor modality (i.e., either SAR or Sentinel-2) from SEN12MS (Schmitt et al., 2019). We assess the performance of these models on the BEN (Sumbul et al., 2019) and SEN12MS-CR (Ebel et al., 2020), employing both sensors individually and in combination. Figure 3 highlights two advantages of our model: (1)

The multisensor pretraining model consistently outperforms models pretrained with a single sensor modality, as indicated by superior performance across all columns when the sensor modality is fixed. (2) Using both sensors for tasks like land use classification and cloud removal leads to enhanced performance, demonstrated by higher accuracy across all rows when the pretraining data is fixed.

The second advantage can be credited to the complementary data offered by both sensors. For instance, SAR images provide crucial information for identifying water bodies and urban structures, as they capture the radar backscatter properties of the Earth's surface. This unique data, when paired with spectral information from Sentinel-2 images, improves classification performance. Moreover, SAR's ability to penetrate clouds significantly contributes to more effective cloud removal results (Xu et al., 2022b).

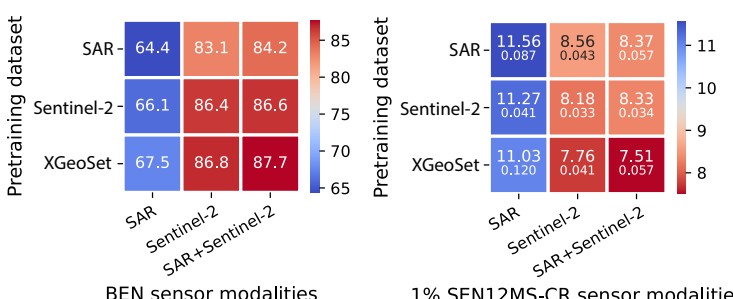

Figure 3: Comparison of our multisensor with single modality pretraining on 10% BEN (left) using mAP (↑) and 1% SEN12MS-CR (right) using SAM (↓). Given the reduced amount of data used for cloud removal (1% of SEN12MS-CR), we conduct the experiment in three replicates and report both mean (top line in each cell) and standard deviations (bottom line in each cell).

Notably, when evaluating the BEN dataset with only the Sentinel-2 modality, our method still achieves a better result than other pretraining methods (86.8%). It rules out the possibility that the improvement of XGeo is only because of sensor modality increase in the downstream tasks.

### 4.3 PRETRAINING FROM SCRATCH PERFORMS BETTER.

Table 5: Distillation from other pretraining model vs pretraining from scatch

| Methods | 1% BEN mAP (↑) | 10% BEN mAP (↑) | SEN12MS-CR MAE (↓) | SAM (↓) | SSIM (↑) | SpaceNet PSNR (↑) | SSIM (↑) | Vaihingen mIOU (↑) |
|---|---|---|---|---|---|---|---|---|
| Distilled from ImageNet22K (Deng et al., 2009) | 79.4 | 86.4 | 0.035 | 6.42 | 0.726 | 22.107 | 0.621 | 72.9 |
| Distilled from CLIP (Radford et al., 2021) | 76.6 | 83.8 | 0.051 | 8.96 | 0.707 | 22.559 | 0.674 | 69.3 |
| Reconstruct CLIP (EVA (Fang et al., 2022)) | 73.5 | 80.6 | 0.053 | 9.96 | 0.689 | 21.778 | 0.591 | 65.7 |
| From scratch | **80.9** | **87.2** | **0.026** | **5.04** | **0.821** | **22.742** | **0.677** | **74.8** |

The effectiveness of using existing vision pretrained models for multisensor geospatial pretraining is evaluated. Intermediate features are extracted (Mendieta et al., 2023) and compared to embeddings from ImageNet-22k. A similar experiment is conducted with the CLIP model (Radford et al., 2021), recognized for its potent multimodal representation learning capabilities. Distillation from ImageNet-22k outperforms that from CLIP (Radford et al., 2021). The EVA method (Fang et al., 2022), which reconstructs the CLIP features of masked patches instead of the patches themselves, is also examined. Despite its proven advantage over traditional MIM like MAE (He et al., 2021a), it surprisingly underperforms in our downstream evaluation compared to other methods, suggesting a larger domain gap for CLIP features (Radford et al., 2021) when applied to multisensor geospatial data.

On the contrary, optimal accuracy for multisensor geospatial pretraining is achieved when a model is trained from scratch. The subpar performance of distillation is credited to the pronounced domain gap between natural images and geospatial-specific sensors. Moreover, distillation has an inherent limitation, as it bounds the student model's performance to align with that of the teacher model (Mendieta et al., 2023). This significant domain gap can be attributed to the fundamental disparities in the physical mechanisms of optical and microwave remote sensing: While optical remote sensing hinges on the reflection and absorption of electromagnetic radiation, microwave remote sensing is governed by scattering, penetration, and dipole-interference of microwaves (Fornaro & Pascazio, 2014). Given that natural images are predominantly underpinned by optical sensors, this results in a

substantial domain discrepancy. Accordingly, for fine-tuning a model for multisensor geospatial tasks, it's advisable to use pretrained weights derived from multisensor data for optimal performance. This finding underscores the need for robust foundation models specific to the geospatial domain, capable of handling diverse sensor data and enhancing performance on multisensor downstream tasks.

## 4.4 ABLATION STUDIES

Table 6: Quantitative results of XGeo, when using dataset homogenous and batch heterogeneous approaches.

| Methods | 1% BEN mAP ($\uparrow$) | 10% BEN mAP ($\uparrow$) | SEN12MS-CR | | | SpaceNet | | Vaihingen |
|---|---|---|---|---|---|---|---|---|
| | | | MAE ($\downarrow$) | SAM ($\downarrow$) | SSIM ($\uparrow$) | PSNR ($\uparrow$) | SSIM ($\uparrow$) | mIOU ($\uparrow$) |
| Dataset homogenous | 77.2 | 84.4 | 0.035 | 10.4 | 0.703 | 19.234 | 0.589 | 72.3 |
| Batch Heterogeneous (Aghajanyan et al., 2021) | **80.8** | **87.2** | **0.026** | **5.04** | **0.821** | **22.742** | **0.677** | **74.8** |

Table 7: Quantitative results of XGeo, with and without MoE/cross sensor reconstruction.

| Pretraining strategies MoE | cross-sensor | Cross sensor percentage | 1% BEN mAP ($\uparrow$) | 10% BEN mAP ($\uparrow$) | SEN12MS-CR | | | SpaceNet | | Vaihingen |
|---|---|---|---|---|---|---|---|---|---|---|
| | | | | | MAE ($\downarrow$) | SAM ($\downarrow$) | SSIM ($\uparrow$) | PSNR ($\uparrow$) | SSIM ($\uparrow$) | mIOU ($\uparrow$) |
| ✗ | ✗ | 0% | 78.3 | 86.2 | 0.038 | 8.19 | 0.735 | 22.333 | 0.589 | 72.8 |
| ✓ | ✗ | 0% | 78.5 | 86.2 | **0.026** | 5.11 | 0.767 | 22.528 | 0.637 | 73.4 |
| ✗ | ✓ | 50% | 80.7 | 86.9 | 0.036 | 8.67 | 0.753 | 22.518 | 0.611 | 73.6 |
| ✓ | ✓ | 100% | 80.5 | 86.8 | **0.026** | **4.96** | 0.789 | 22.634 | 0.649 | 74.4 |
| ✓ | ✓ | 50% | **80.9** | **87.5** | **0.026** | 5.04 | **0.821** | **22.742** | **0.677** | **74.8** |

**Heterogeneous Batching.** Another critical factor in achieving successful multi-sensor pretraining and learning generalizable representations is the selection of batches. Inspired by multitasking learning (Aghajanyan et al., 2021). We experimented with two balancing schemes: dataset homogenous and batch heterogeneous (Aghajanyan et al., 2021). As the result shown in Table 6, heterogeneous batching shows a superior performance than dataset homogenous. It demonstrates that heterogeneous batching not only works in the multitasking prefineutning, but also significantly impacts the performance and generalizability of a multi-sensor pretrained model.

**MoE and cross sensor pretraining.** In the proposed XGeo model, we incorporate both cross-sensor pretraining paradigms and the Mixture of Experts (MoE). In an ablation study, we present the results when either MoE or cross-sensor pretraining is omitted. As shown in Table 7, removing MoE from the model results in similar performance on the BEN dataset, while other tasks see a more substantial decrease. This uneven response across different tasks aligns with observations made in several previous multi-modal studies (Zhu et al., 2022). On the other hand, removing the cross-sensor paradigm leads to a consistent performance decline across all tasks.

It is natural to question how the proportion of sensor crossing affects performance. To explore this, we perform an ablation study on the percentage of sensors subject to cross-reconstruction. Our results suggest that a sensor crossing rate of 50% provides slightly superior outcomes compared to a rate of 100%. This indicates that the optimal sensor crossing strategy maintains a balance between the benefits of cross-reconstruction and the retention of sensor-specific information, consequently enhancing performance across a diverse range of geospatial tasks.

## 5 CONCLUSION

We present a multisensor pretraining model that incorporates a novel cross-sensor paradigm for joint representation learning, effectively capturing relationships between corresponding sensors. To handle the diverse representation of geospatial data during pretraining, we employ a MoE and heterogeneous batching. Our pretrained model is based on a large-scale multisensor dataset comprising over 2 million images. Our approach demonstrates superior performance across various multisensor downstream tasks. Limitation and future direction of our study can be found in Appendix D

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

# A  OPTIMIZATION ON PRETRAINING DATA

## A.1  PERFORMANCE WITH OTHER CHOICE OF PRETRAINING DATA.

To optimize XGeoSet, we initially focused on optimizing XGeoSet-RGB. As previous research has indicated, a successful pretraining dataset requires rigorous testing of each component (Nguyen et al., 2023). Thus, we conducted a series of experiments on each individual dataset. These experiments involved the use of ImageNet (Deng et al., 2009) with 3 million images, GeoLifeCLEF with 3.3 million images, and the Functional Map of the World (FMoW) (Christie et al., 2018). For FMoW, we segmented the dataset into tiles of size 384, leading to a total of 6 million images. This diverse selection of datasets allowed us to comprehensively test and optimize our pretraining approach for XGeoSet.

To ensure other variables, such as the backbone architecture and pretraining methodologies, do not skew our results, we chose to employ the Swin-base(Liu et al., 2021) and committed to pretraining from scratch. In line with our aim for equitable comparison, we also adhered to the same seven downstream tasks as delineated in the previous report (Mendieta et al., 2023). This approach creates a consistent testing environment across all datasets, reducing the potential for bias or error.

Interestingly, upon integrating the GeoLifeCLEF into our testing framework, we observed a downturn in performance on downstream tasks. This result signifies that not all datasets necessarily contribute to improved model performance, and their selection demands careful consideration.

Even though the addition of both the Functional Map of the World dataset and ImageNet gave rise to performance metrics that were commensurate with those achieved by XGeoSet-RGB, these new dataset additions were not as efficient. The key reason for this inefficiency was the significantly larger size of the pretraining dataset, which introduced higher computational costs and longer processing times. This finding highlights the importance of carefully balancing dataset size and complexity with computational efficiency in the model training process.

Table 8: Results of downstream tasks with different pretraining datasets: change detection and classification

| Dataset | # Image | OSCD (F1) | DSFIN (F1) | BEN 10% | BEN 1% |
|---|---|---|---|---|---|
| GeoPile (Mendieta et al., 2023) | 600K | **57.5** | 66.2 | 86.4 | 79.3 |
| XGeoSet-RGB | 1.7M | 57.1 | **70.4** | **86.8** | **79.6** |
| XGeoSet-RGB + ImageNet (Deng et al., 2009) | 3M | **57.5** | 69.2 | 86.4 | 79.5 |
| XGeoSet-RGB + GeoLifeCLEF | 3.3M | 56.1 | 61.6 | 86.1 | 78.9 |
| XGeoSet-RGB + FMoW (Christie et al., 2018) | 6M | 58.2 | 69.3 | 86.2 | 79.1 |

Table 9: Results of downstream tasks with different pretraining datasets: segmentation and super-resolution

| Dataset | # Image | WHU | Vai. | SN2 (PSNR) | SN2 (SSIM) |
|---|---|---|---|---|---|
| GeoPile (Mendieta et al., 2023) | 600K | 90.1 | 75.1 | **22.626** | 0.645 |
| XGeoSet-RGB | 1.7M | **90.6** | 75.9 | 22.599 | **0.658** |
| XGeoSet-RGB + ImageNet (Deng et al., 2009) | 3M | 90.5 | **76.1** | 22.107 | 0.631 |
| XGeoSet-RGB + GeoLifeCLEF | 3.3M | 89.1 | 74 | 16.663 | 0.512 |
| XGeoSet-RGB + FMoW (Christie et al., 2018) | 6M | 90.2 | 75.7 | 22.448 | 0.638 |

## A.2 Performance without RGB modality

RGB modalities are singled out because of the abundance of RGB datasets that come from various sources beyond just Sentinel-2. For instance, MillionAID (Long et al., 2021), a dataset comprised of a wide range of RGB images, is sourced from multiple satellites, including GeoEye, WorldView, QuickBird, IKONOS, and SPOT satellites, among others. Additionally, a previous study (Cong et al., 2022) found that using only Sentinel-2 data for pretraining does not yield optimal performance in the downstream evaluation. Therefore, we sought to diversify our sources and include a wider range of RGB images in our pretraining data. This breadth of data sources significantly enriches the diversity of the RGB modality in our study.

Despite overlapping GSD in some RGB modality, more geospatial features will be included. Although these datasets may not provide an imaging spectrum as wide as Sentinel-2, they enhance the entropy of pre-training data, which has been proven to be effective in (Mendieta et al., 2023), which is demonstrated by Table 10.

Table 10: Results pretrained with single modality or without RGB modality.

| Pretraining sensor modality | 10% BEN | cloud removal |
|---|---|---|
| Metric | mAP ($\uparrow$) | SAM ($\downarrow$) |
| SAR (in Figure 3) | 84.2 | $8.37 \pm 0.057$ |
| Sentinel-2 (in Figure 3) | 86.6 | $8.33 \pm 0.034$ |
| RGB | 86.4 | $10.45 \pm 0.12$ |
| w/o RGB | 86.6 | $9.67 \pm 0.12$ |
| XGeoSet | **87.7** | $\mathbf{7.51} \pm 0.057$ |

## B Pretraining Details

### B.1 Pretraining Settings

**Masking.** All hyper-parameters are listed in Table 11. We implement a masking strategy that maintains consistency around different channels within the same sensor, applying the mask at the same locations. However, when it comes to different sensors, we employ a varying masking approach, ensuring that the mask is applied at different locations. This methodology allows us to preserve sensor-specific information while investigating inter-sensor discrepancies effectively.

**Heterogeneous batch size.** Given the disparity in the number of images obtained from different sensors, we employ a heterogeneous batch size strategy for our training process. This methodology adjusts the batch size in proportion to the amount of data sourced from each individual sensor. In essence, during each epoch of our training process, every type of sensor is iterated through once, irrespective of the data volume associated with that particular sensor. This ensures that all sensor types have an equal chance to contribute to the model's learning process, fostering a more balanced and comprehensive training regimen. Alongside this, we also adjust the learning rate proportionally in accordance with the batch size allocated per sensor.

**Cross sensor pretraining.** During the pretraining phase, in instances where sensors are colocated, we utilize a cross-sensor pretraining approach. This methodology allows for a possibility of either

Table 11: Hyperparameters of XGeo pretraining.

| Hyper-parameter | Value |
|---|---|
| Backbone | SwinV2-Base (Liu et al., 2022b) |
| Image size | $192 \times 192$ |
| Optimizer | AdamW |
| $\beta_1$ | 0.9 |
| $\beta_2$ | 0.999 |
| Eps | $1.0 \times 10^{-8}$ |
| Momentum | 0.9 |
| Weight decay | 0.05 |
| Learning rate | $\{1.0 \times 10^{-4}, 0.25 \times 10^{-4}, 1.0 \times 10^{-5}\}$ for RGB, Sen12MS (Schmitt et al., 2019) and MDAS (Hu et al., 2023) |
| Warm up learning rate | $5.0 \times 10^{-7}$ |
| Weight decay | $10^{-5}$ |
| Batch size | $\{128, 32, 12\}$ per GPU for RGB, Sen12MS (Schmitt et al., 2019) and MDAS (Hu et al., 2023) |
| Training epochs | 800 or 100 |
| Warm up epochs | 10 |
| Learning rate decay | Multistep |
| Gamma | 0.1 |
| Multisteps | [700,] for 800 or [] for 100 |
| # Experts | 8 |
| MoE blocks | 1, 3, 5, 7, 9, 11, 13, 15, 17 (Every other swin block) |
| Top-value ($k$) | 1 |
| Capacity factor | 1.25 |
| Aux loss weight ($\lambda$) | 0.01 |
| Mask patch size | 32 |
| Mask ratio | 0.6 |

self-reconstruction or reconstruction of images from colocated sensors. This dual reconstruction strategy ensures a broader perspective and a richer learning experience for the model.

### B.2 VISUALIZATION

To demonstrate this methodology, we provide several examples in Figure 4. These instances visually illustrate our cross-sensor pretraining approach, highlighting the ability of RGB to self-reconstruct, as well as the excellent mutual reconstruction capabilities between DSM and RGB images. However, self-reconstruction and mutual reconstruction involving SAR images poses some challenges, as we use unprocessed, noisy SAR images. Due to the structure of MIM, which involves an encoder and a lightly constructed decoder, only low-frequency components of the images are reconstructed Xie et al. (2021); He et al. (2021a), making the SAR reconstruction slightly difficult. Specifically, we visualized the statistics (Bauer-Marschallinger et al., 2021) and SSI (Sheng & Xia, 1996) value before and after the reconstruction for both HV and VV bands (Figure 5).

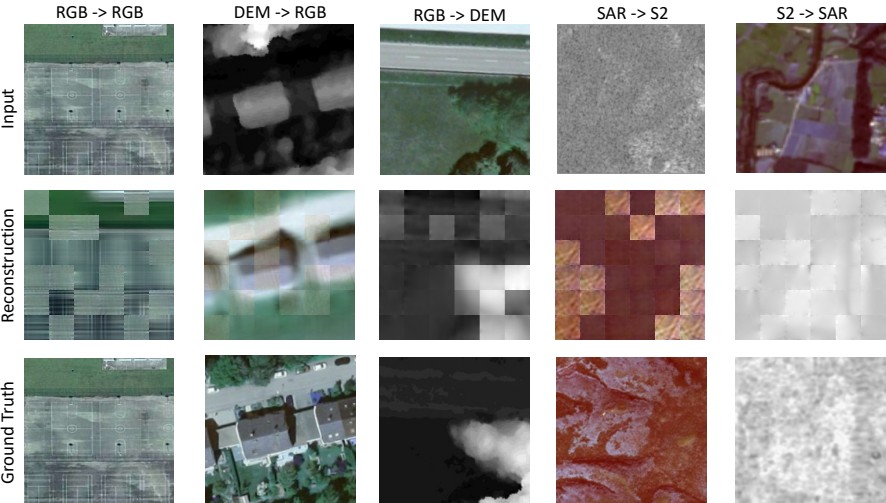

Figure 4: Examples of cross-sensor pretraining. We note that the first row represents input before masking. Second row represents reconstruction. Third row represents the reconstruction groundtruth.

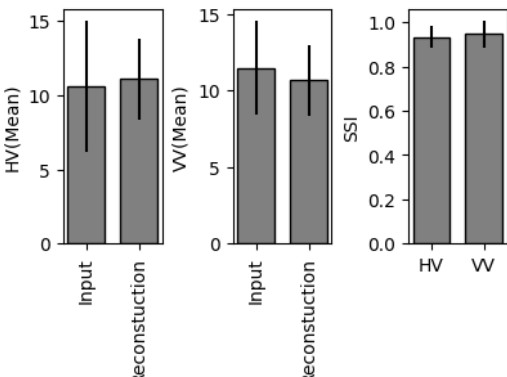

Figure 5: Figure R1: SAR backscatter statistics comparing input and reconstruction using the MIM. The two bands of SAR are HV and VV. The mean and standard deviation for the HV band are shown on the left, while those for the VV band are displayed on the right. The Speckle Suppression Index (SSI) values are presented in the right panel. An SSI value closer to one indicates that the mean and standard deviation remain consistent before and after reconstruction.

## C  DOWNSTREAM EXPERIMENTS

### C.1  MODEL SIZE

Regarding the number of parameters, we followed a standard backbone for pretraining, the details of which have been reported in Xie et al. (2021). Comparisons between training from scratch and using ImageNet pretrained weights have been provided in our tables and corroborated by previous studies (Mañas et al., 2021; Cong et al., 2022; Chen et al., 2020; Wanyan et al., 2023; Mendieta et al., 2023).

Table 12: Model size

| Model | SeCo | SatMAE | MoCoV2 | DINO-MC | GFM | XGeo |
|---|---|---|---|---|---|---|
| # of trainable parameters | 23M | 307M | 23M | 48.6M | 89M | 89M |

## C.2 EXPERIMENTAL SETTINGS

There are primarily two ways to leverage pretrained weights, as depicted in Figure 6. The first approach involves feeding each sensor through encoders that share weights. The resulting embeddings are then concatenated and fed into the classifier. In the second approach, all sensor data are stacked together in the color channel prior to patchification. This approach resembles the multiMAE method Bachmann et al. (2022), where the projected patches from all modalities are concatenated into a single sequence. Our experiments on 1% of the BEN dataset Sumbul et al. (2019), listed in Table 13, demonstrate that both methods yield comparable results. However, the latter approach is more computationally efficient. Therefore, all results mentioned in the main text utilize this second approach. Importantly, in both cases, no masking is performed during the transfer phase.

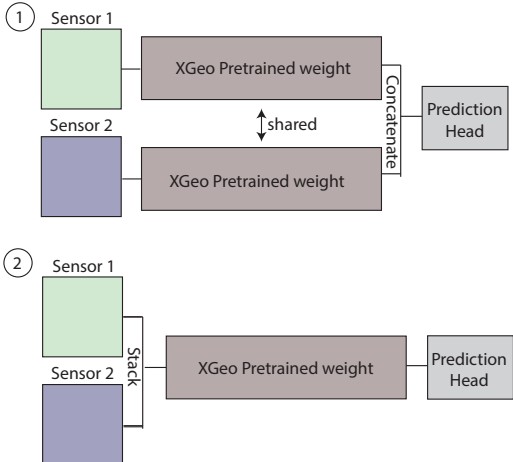

Figure 6: Two methods of downstream transfer. In the top panel, every sensor is fed into a separate encoder initialized with XGeo pretrained weight. The embeddings from the last layer are concatenated, and then fed through the prediction head, such as classifier and segmentation decoder. In lower panel, images are concatenated along the color channel and then fed through one encoder initialized with XGeo pretrained weight.

Table 13: Results of BEN when comparing different downstream transfer methods illustrated in Figure 6

| Finetuning Method | BEN 1% |
| --- | --- |
| 1 | 80.8 |
| 2 | 80.8 |

**Cloud removal.** For tasks associated with the SEN12MS-CR dataset Ebel et al. (2020), we employ a standard decoder, as delineated in the SimMIM study Xie et al. (2021), to recover the original input size from the features extracted by the encoder. We retain the SimMIM parameters for the optimizer, weight decay, and other training settings to maintain consistency with the original study Xie et al. (2021). However, we deviate from the initial study by not incorporating any random augmentations. Our training protocol employs a batch size of 64, with each GPU assigned 16 images, and the dimensions of the input images are fixed at 160×160 pixels. The training is conducted over 100 epochs, with an initial learning rate set at 1.25e-4, similar to other tasks.

**Classification and segmenation.** Other tasks follow exactly as Mendieta et al. (2023).

## C.3 PERFORMANCE WITH LESS TRAINING DATA.

Table 14 presents the results from the 1% Big Earth Net presented in tables 5-7 from the main text, assembling few-show. We also further reduce the number of examples per class, shown below.

Table 14: Results for 1% Big Earth Net and average 5 examples per class

| Finetuning Method | 1% BEN (duplicated from the paper) | Average 5 examples per class |
|---|---|---|
| SeCo (Mañas et al., 2021) | 63.6 | 32.5 |
| SatMAE (Cong et al., 2022) | 68.9 | 35.9 |
| GFM (Mendieta et al., 2023) | 80.7 | 37.4 |
| Random | 65.8 | 34.5 |
| ImageNet (Deng et al., 2009) | 79.4 | 37.5 |
| XGeo | **80.9** | **38.8** |

## C.4 DISCUSSIONS OF MORE DOWNSTREAM TASKS.

**Pan-sharpening.** Pansharpening, akin to super-resolution, involves the combining a high-resolution grayscale panchromatic image with the color information from a low-resolution multispectral image to generate a high-resolution color image. For this assessment, we utilized the SpaceNet2 dataset, following the methods in (Mendieta et al., 2023). We juxtaposed the performance of our model with a series of baseline models, measuring the outcomes using peak signal-to-noise ratio (PSNR) and SSIM. As illustrated in Table 3 in the main text, our model demonstrates superior performance over its competitors. Notably, the SpaceNet dataset comprises images from the Worldview 3 satellite, a source absent from our pretraining data, demonstrating a good transferability of our pretrained model across diverse sensors.

**Segmentation.** Segmentation is another popular remote sensing application for enabling automated extraction of building footprints or land cover mappings over wide regions. We therefore conduct experiments on this task on two different datasets. Vaihingen (Rottensteiner et al., 2012) is an urban semantic segmentation dataset collected over Vaihingen, Germany at a GSD of 0.9m. The experiment settings are the same as (Mendieta et al., 2023). We report the intersect of union (IoU) segmentation results for all methods in Table 3 in Main text. Our approach is able to provide the best result.

## C.5 QUALITATIVE EXAMPLES

We present some quantitative results of cloud removal and segmentation in Figure 7 and Figure 8 respectively.

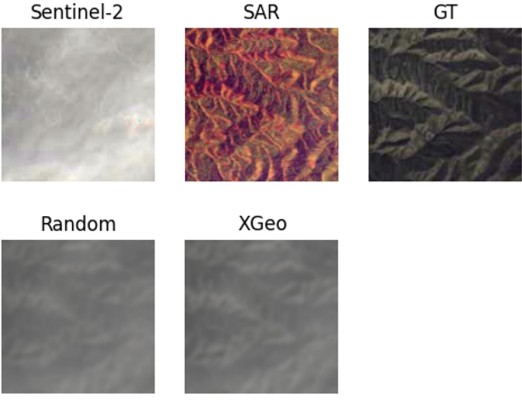

Figure 7: A display of qualitative results of the cloud removal from XGeo in comparison to other competitive methods.

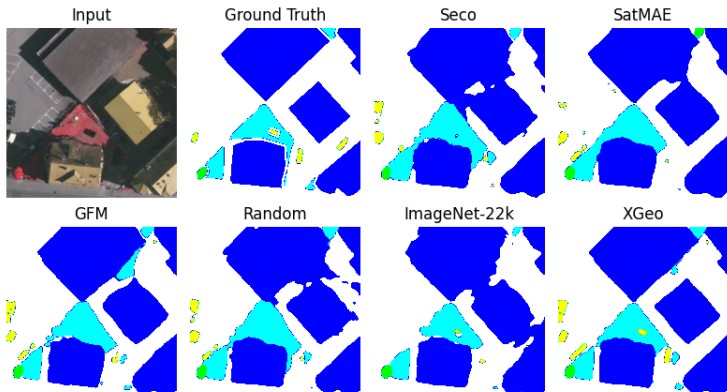

Figure 8: A display of qualitative results showcasing segmentation outcomes from XGeo in comparison to other competitive methods.

## D  LIMITATIONS, FUTURE DIRECTION

A potential limitation of our approach lies in the suboptimal incorporation of multitemporal data into our pretraining dataset. While SatMAE (Cong et al., 2022) introduces a temporal encoding, it has yet to demonstrate effective handling of tasks requiring temporal information. While the inclusion of temporal information is undoubtedly valuable and could potentially strengthen the model, it also poses significant challenges. The major issue is the exponential increase in pre-training costs that incorporating temporal information would entail. Therefore, effectively integrating spatial and temporal information in a pretrained model goes beyond simply incorporating the data - it requires substantial methodology design.

In future work, we aim to enhance our model's capability to support downstream tasks where temporal information is crucial, such as ecosystem prediction. Furthermore, we only present pretraining dataset where two sensors are colocalized and sensors with no colocalization, it's worth noting that our model design could accommodate pretraining with more colocalized sensors.

