# OpenReview forum: "Multisensory Geospatial Models via Cross-Sensor Pretraining"
_ICLR.cc/2024/Conference — ICLR 2024 Conference Withdrawn Submission_

### Official Review · Reviewer_V45f · 2023-10-29

**Soundness:** 3 good
**Presentation:** 3 good
**Contribution:** 2 fair
**Rating:** 6
**Confidence:** 4

**Summary:**

This paper introduces a pretraining methodology based on MAE for multi-sensor
data, specifically with RGB data, 14-channel Sentinel 2 data, 2-channel
synthetic aperture radar (SAR), and digital surface models (DSM).

Different sensors are given different tokenizing encoder layers, and different
decoder layers, but all sensors share the same Swin backbone.

Multiple downstream tasks (land-use classification, segmentation, cloud
segmentation, pan-sharpening) are evaluated.

Heterogeneity of the data is mitigated with heterogeneous batches and
per-sensor encoders.

XGeoSet contains 2 million images and is a union of 4 other datasets.

Experiments demonstrate performance increases on a range of downstream tasks.

**Strengths:**

* The research questions are clearly laid out.

* Evaluation is well thought out and comprehensive.

**Weaknesses:**

* Evaluation results do not contain error bars, which makes it difficult to
  compare pointwise results.

* A large part of the paper is in the Appendix. This is a strength and a
  weakness, the studies are comprehensive, but it makes scoping difficult in
  the context of a conference.

**Questions:**

> XGeoSet, encompassing over 2 million images.

You should state the size of the images (on average if they are heterogeneous).
2 million 1024x1024 images is a lot different than 2 million 128x128 images.


> Paired data in the abstract.

This could use clarification. I think this means you could have 2 or more
sensor readings for the same spacetime location?

> In Section 4: When specified, 1% BEN and 1% SEN12MS-CR are also employed for ablation studies.

It is unclear what 1% means in this context. Also BEN is undefined at this
point (it is defined later in section 4.1) but it should be defined on first
usage.


> The models are pre-trained for either 100 epochs for ablation studies or 800 epochs

Can you be confident that trends wont change down the line?


> Table 5: Distillation from other pretraining model vs pretraining from scatch

Could the result that from-scratch works better be due to not spending as much time tuning the other methods?

---

### Official Review · Reviewer_KKqz · 2023-10-30

**Soundness:** 3 good
**Presentation:** 3 good
**Contribution:** 1 poor
**Rating:** 3
**Confidence:** 5

**Summary:**

The submitted manuscript introduced a method for "cross-sensor" pretraining of geospatial data. The underlying transformer architecture uses a shared encoder with a mixture-of-experts (MOE) setup to capture dependencies between different input sensors. Pretrianing is done as masked image modelling on the following modalities: RGB, Sentinel-2, Sentinel-1, and a Digital Surface model (DSM). Results demonstrate a minor performance gain for some downstream tasks and larger outperformance for some other downstream tasks as compared to baselines and SOTA.

This submission introduces indeed an interesting approach for the remote sensing community. I am not so sure about the ICLR community and its focus on methodological novelty. The method proposed uses known ML or computer vision methods applied on remote sensing data.

**Strengths:**

The submission has the following strength:

**#1.** easy to read and follow

**#2.** is well-structured

**#3.** present an novel combination of known approaches to the interesting domain of remote sensing data

**Weaknesses:**

The submission has the following weaknesses:

**#1.** As already mentioned in the summary, this work presents little methodological novelty and rather applies known methods from ML/CV to the remote sensing domain. There is nothing wrong with this, but I think that this work would be a much better match for a remote sensing journal than ICLR.

**#2.** This work claims "to our best knowledge, it is the first multi-sensor geospatial pretraining of such kind" i.e., able to deal with paired and unpaired data. Then it reports related work such as Liu et al. (2022a), Chen & Bruzzone (2022) and Scheibenreif et al. (2022) doing pretrained with paired data from Sentinel-2 and Sentinel-1. Why is there no comparison with this related work presented (pretrained models of these works are available)? As far as it goes for "unpaired" data, I can not see any ablation study showing the method's capability to outperform on "unpaired" data. I am asking because the overall motivation of this work is given by Fig. 1. which is showing "paired" image modalities. Maybe I missed it and as far as I understand "Section 4.2 COMPARISON WITH SINGLE SENSOR PRETRAINING", the authors compare results on paired pretraining, not explicitly pretraining on "unpaired" data. Again, in Section 4.2, I would love to see comparisons against the related work mentioned above.

**#3.** I have difficulties understanding the expression: "...we have made discoveries, yet to be reported...initiating pretraining from scratch has been observed to yield superior results compared to leveraging existing foundational models". I am not sure what the authors mean by this? Do they compare to pretrianing from other geospatial models, pretraining from ImageNet or just training from scratch?

**#4.** It would be great to see how this method is able to perform beyond remote sensing data. The approach might be applicable on other domains where multiple modalities are present such as e.g., medical images or other sensor fusion setups.

**#5.** Unfortunately, the reported performance of the presented method only marginally outperforms GFM (Mendieta et al., 2023). Looking at Table 3, it would be great to see GFM performance at the "SEN12MS-CR" downstream to better compare both models.

**Questions:**

I have the following question:

**#1.** Is there any ablation study showing the performance of the proposed model for "unpaired" data sources? Maybe the authors know a setup, which could demonstrate the proposed methods capability to work on "unpaired" data.

---

### Official Review · Reviewer_Tpx5 · 2023-10-31

**Soundness:** 3 good
**Presentation:** 3 good
**Contribution:** 3 good
**Rating:** 8
**Confidence:** 4

**Summary:**

This paper proposes a self-supervised model for multi-modal remote sensing images. Based on masking strategy and cross-modality reconstructions, this foundation model is then used for different benchmark tasks.

**Strengths:**

The paper is well written, the results seems good on the various downstream tasks, and the model architecture and training is relevant. This foundation model could be useful for the community.

**Weaknesses:**

It is not clearly indicated whether the codes and the pre-trained models will be accessible after publication or not.

I would need a clarification. While the model seems to be able to process up to 4 modalities, it looks as if only 2 are simultaneously used, either during pre-training or during downstream tasks. Would it be significantly different to just have 2 different models with each 2 modalities? Was it not possible to find a dataset or task using 3 modalities?

**Questions:**

All images seems to be aligned (co-registered), yet for example SAR images are initially in radar geometry and the projection leads to data loss and artifacts.  Do you think it would be possible to use (I mean re-train) your model even if the images are in different geometries?

Why 8 experts?

Please explain better how to go from pre-training to downstream tasks: what are the pieces that are used, what is added, what is learned, etc

Section 3.2: How does the 'either' (reconstructing itself or its paired sensor) is implemented in practice? is it random? or always both are performed?

Typos/smaller remarks:
'geospaital'
define MoE in part 1
'methodology. .'
'or' --> random? or both?
'Section ??'

---

### Official Review · Reviewer_2AtA · 2023-11-07

**Soundness:** 2 fair
**Presentation:** 3 good
**Contribution:** 3 good
**Rating:** 3
**Confidence:** 4

**Summary:**

This work considers the problem of pretraining computer vision models for multimodal remote sensing data. The proposed method is based on masked autoencoders (MAEs). The basic idea is to perform cross-modal reconstruction (with some probability, when paired modalities are available) and traditional input reconstruction otherwise. This allows for pretraining on heterogeneous datasets which may include some locations that are imaged with multiple modalities and other locations which are only imaged with a single modality. All modalities have separate patch embedding layers and decoders, while the encoder is shared. The pretrained representations are evaluated on a number of downstream geospatial tasks.

# References

@inproceedings{jean2019tile2vec,
  title={Tile2vec: Unsupervised representation learning for spatially distributed data},
  author={Jean, Neal and Wang, Sherrie and Samar, Anshul and Azzari, George and Lobell, David and Ermon, Stefano},
  booktitle={Proceedings of the AAAI Conference on Artificial Intelligence},
  volume={33},
  number={01},
  pages={3967--3974},
  year={2019}
}

**Strengths:**

* The paper considers an important question.
* Integrating information from multiple aligned modalities is a reasonable thing to do, and the proposed approach is intuitive and sensible. Broadly, I like the proposed approach.
* The proposed method is evaluated on a number of reasonable remote sensing domain tasks.
* The paper includes sensible ablation studies.
* The paper claims to be the first self-supervised method in the geospatial domain that can handle paired and unpaired multimodal data. Based on a quick search, I didn't find obvious counterexamples. Perhaps someone with more domain experience can weigh in.
* Full hyperparameters are specified in the appendix.
* Qualitative results included in appendix.

**Weaknesses:**

# Experiment issues
* As the paper points out, the results in Table 3 are "replicated from previous reports" and they are based on a variety of backbone architectures. This makes it very difficult to draw any conclusions about the proposed method from Table 3.
* Based on section 3.3, it sounds like pretraining datasets were chosen to maximize the performance of the proposed method. This does not seem to be the makings of a fair comparison...
* The stability of the method/results under retraining is not clear.
* The results in Figure 3 seem like they would be confounded by dataset size. The advantage of SAR seems slight - couldn't it just be a regularization effect?
* How were hyperparameters tuned?
* All of the datasets in this paper vary in resolution, and this doesn't seem to be controlled for anywhere - isn't this a confounder for some of the experiments in the paper? Couldn't some of the performance differences be due to similarities between the resolution of pretraining and downstream tasks, as opposed to other factors like the modality? This may be relevant for e.g. Table 8, 9, and 10.

# Other issues
* Some important related work on pretraining in the geospatial domain is missing, e.g. [jean2019tile2vec]. Please double check to make sure other important works were not omitted.

**Questions:**

* See "Experiment issues".
* Given the information under "Experiment issues", does the paper make a clear case that the proposed method provides a stable and substantial advantage over existing alternatives?

# Misc. Comments
* There are two periods at the end of the first paragraph in section 3.
* Why does it make sense to have per-sensor embeddings instead of per-channel embeddings? Different bands can differ substantially, and can be considered different modalities.
* I would like to see a discussion of scalability as the number of sensors increases.
* Shouldn't $C_m$ be $C_e$ in section 3.2?
* Is it appropriate to give a new name ("XGeoSet") to the collection of 4 (pre-existing) datasets that are used for pretraining? It doesn't seem necessary to me.
* Have there been any attempts to balance pretraining across datasets or resolutions? Might that help?
* Are there some results about which the cross-modal losses? Which modality pairs are possible to cross-predict, and which are not? For instance, DSM->RGB seems quite difficult.
* Broken reference in section 3.4 ("??")
* The discussion of batch size is confusing - section 3.4 says that batch can be depicted as a set indexed by $i \in \{1, \ldots, N\}$ where $N$ is the number of sensors. This implies that the batch size is $N$. But later the batch size is stated as 2048.
* Please include a citation for GeoLifeCLEF - which year's version was used?